# Every sip counts: Understanding hydration behaviors and user-acceptability of digital tools to promote adequate intake during early and late pregnancy

Abigail M. Pauley[1]*, Asher Y. Rosinger[2,3], Jennifer S. Savage[4], David E. Conroy[1], Danielle Symons Downs[1,5]

1 Department of Kinesiology, The Pennsylvania State University, University Park, Pennsylvania, United States of America, 2 Department of Biobehavioral Health, The Pennsylvania State University, University Park, Pennsylvania, United States of America, 3 Department of Anthropology, The Pennsylvania State University, University Park, Pennsylvania, United States of America, 4 Department of Nutritional Science and Center for Childhood Obesity Research, The Pennsylvania State University, University Park, Pennsylvania, United States of America, 5 Department of Obstetrics and Gynecology, Penn State Health Milton S. Hershey Medical Center, Hershey, Pennsylvania, United States of America

* amp34@psu.edu

**Data Availability Statement:** Data cannot be shared publicly because the data collected on human subjects was not IRB approved to be

## Abstract

Maintaining adequate hydration over the course of pregnancy is critical for maternal and fetal health and reducing risks for adverse pregnancy outcomes (e.g., preeclampsia, low placental and amniotic fluid volume). Recent evidence suggests that women may be at risk for under-hydration in the second and third trimesters when water needs begin to increase. Scant research has examined pregnant women's knowledge of hydration recommendations, water intake behaviors, and willingness to use digital tools to promote water intake. This study aimed to: 1) describe hydration recommendation knowledge and behaviors by the overall sample and early vs late pregnancy, and 2) identify habits and barriers of using digital tools. Pregnant women ($N$ = 137; $M$ age = 30.9 years; $M$ gestational age = 20.9) completed a one-time, 45-minute online survey. Descriptive statistics quantified women's knowledge of hydration recommendations, behaviors, and attitudes about utilizing digital tools to promote adequate intake, and Mann-Whitney U and chi-squared tests were used to determine group differences. Most women lacked knowledge of and were not meeting hydration recommendations (63%, 67%, respectively) and were not tracking their fluid consumption (59%). Knowledge of hydration recommendations differed by time of pregnancy, such that women in later pregnancy reported 82 ounces compared to women in early pregnancy (49 ounces). Common barriers included: forgetting to drink (47%), not feeling thirsty (47%), and increased urination (33%). Most were willing to use digital tools (69%) and believed a smart water bottle would help them achieve daily fluid recommendations (67%). These initial findings suggest that pregnant women may benefit from useful strategies to increase knowledge, decrease barriers, and maintain adequate hydration, specifically earlier in pregnancy. These findings will inform the design of a behavioral intervention incorporating smart connected water bottles, wearables for gesture detection, and behavior modification strategies

shared publicly. Also, since it is human subjects data, a data transfer agreement is needed. Data are available from the Pennsylvania State University Office for Research Protections at orp@psu.edu or 814-865-1775 for researchers who meet the criteria for access to confidential data.

**Funding:** This project was supported by the Penn State Clinical & Translational Research Institute, Pennsylvania State University CTSA, and NCATS Grant Number UL1 TR000127 and UL1 TR002014. The funders had no role in study design, data collection and analysis, decision to publish, or preparation of the manuscript.

**Competing interests:** The authors have declared that no competing interests exist.

to overcome barriers, promote proper hydration and examine its impact on maternal and infant health outcomes.

## Author summary

Maintaining adequate hydration over the course of pregnancy is critical for maternal and fetal health and reducing risks for adverse pregnancy outcomes (e.g., preeclampsia, low placental and amniotic fluid volume). Recent evidence suggests that women may be at risk for under-hydration in the second and third trimesters when water needs begin to increase. Scant research has examined pregnant women's knowledge of hydration recommendations, water intake behaviors, and willingness to use digital tools to promote water intake. This study aimed to: 1) describe hydration recommendation knowledge and behaviors by the overall sample and early vs late pregnancy, and 2) identify habits and barriers of using digital tools. Initial findings suggest that pregnant women may benefit from useful strategies to increase knowledge, decrease barriers, and maintain adequate hydration, specifically earlier in pregnancy. These findings will inform the design of a behavioral intervention incorporating smart connected water bottles, wearables for gesture detection, and behavior modification strategies to overcome barriers, promote proper hydration and examine its impact on maternal and infant health outcomes.

## Introduction

Maintaining adequate hydration over the course of pregnancy is critical for maternal and fetal health and reducing risks for adverse pregnancy outcomes [1–6]. To ensure proper functioning and growth for both mother and fetus (i.e., adequate amniotic fluid volume and plasma volume balance, proper fetal brain development, and reduced risk of low-birth weight and fetal arterial hypertension [7,8]), hydration needs increase above what is required for non-pregnant individuals. Hydration guidelines suggest that non-pregnant women ages 18–50 years old, should consume 2.7 L/day of total water consumption, which includes 2.2 L (9 cups) as total beverages, including drinking water [7]. Pregnant women should consume 3.0 L/day of total water, which includes approximately 2.3 L (10 cups, ~80 ounces) as total beverages, including drinking water [7]. However, the American College of Obstetricians and Gynecologists recommend a range of 8 to 12 cups/day (64 to 96 ounces/day, 1.9 to 2.8 L/day) [9], which accounts for varied needs, due to greater body composition and body mass, living in hotter temperatures and engaging in physical activity [10,11]. In addition to drinking water, pregnant women are also encouraged to incorporate water-dense foods in their diet (i.e., fruits, vegetables) and track their water intake. If pregnant women are not meeting these guidelines, under-hydration may occur, leading to adverse pregnancy outcomes. New mothers and their fetuses/infants may be at a greater risk for maternal and infant mortality. Proper hydration may reduce the risks of headaches, constipation, low amniotic fluid, inadequate breast milk production, and premature birth [1–7,9].

However, few studies have examined how much water pregnant women are drinking and if they are meeting current hydration recommendations. For example, one study by Bardosono and colleagues [12] examined water intake behaviors and knowledge of hydration recommendations of pregnant women living in Indonesia. They found that 42% of pregnant women fell short of the recommended water intake of 2.08 L/day and 14% lacked knowledge of the water intake recommendations. Another study by Zhou and colleagues [13] examined water intake

in women living in China and found that the majority of pregnant (72%) and breastfeeding (73%) women were not meeting the Chinese Nutrition Society [14] total water intake recommendations of 3.0 L/d and 3.8 L/day, respectively. The Centers for Disease Control and Prevention [15] report that women over 20 years of age are only consuming on average, 1.23 L of water each day, which suggests women who become pregnant are not drinking enough to begin with, which could lead to further under-hydration.

A third study by Rosinger and colleagues [16] found evidence to suggest that pregnant women may be at risk for under-hydration in the second and third trimesters when water needs begin to increase. Rosinger and colleagues [16] examined urine osmolality and under-hydration levels in pregnant women with overweight/obesity participating in the Healthy Mom Zone gestational weight gain regulation intervention [17,18]. As part of a larger, multi-component intervention, women in the intervention group were encouraged to track their water intake, increase intake of water-dense foods, and meet the hydration guidelines of consuming at least 2.4 L of water from ~8–36 weeks gestation [7,17,18]. Women used the MyFitnessPal app (a smartphone app used to track daily food and beverage intake, and exercise behaviors; [19]) to track their daily hydration behaviors and biomarkers of hydration levels (i.e., weekly urine samples) were collected and analyzed to examine hydration levels [16]. These analyses found that the intervention was successful in promoting proper hydration in pregnant women with overweight/obesity [16]. More specifically, the intervention women maintained a significantly lower overnight urine osmolality (adequately hydrated) compared to the control group who had higher overnight urine osmolality during the second and third trimesters [16]. The MyFitnessPal app [19] was used to track diet quality and hydration behaviors as maternal nutrition (eating water-dense foods, drinking water and water-based fluids) is essential to promote healthy pregnancies and offspring health. However, over- and under-reporting of foods and beverages has been widely documented, thus highlighting the need for other more accurate and reliable modes of measurement [20,21]. The development of other digital tools may also offer benefits for promoting hydration.

For example, one digital tool that has many variations, is a smart water bottle, which tracks the volume of water consumed throughout the day and alerts the user when not drinking enough in relation to their personal fluid intake goal, has been developed to improve water intake levels. One study examined the thoughts and preferences of 94 individuals with kidney stones in relation to making lifestyle changes to prevent kidney stones. Streeper and colleagues [22] found that technology has not been previously used in meeting fluid intake recommendations in this population and most individuals felt that an app or device would improve their adherence and would be interested in using an app or device [22]. Based on these findings, Conroy and colleagues [23] developed a Just-in-Time adaptive intervention (an intervention design that aims to provide the right amount/type of support, at the most opportune time, by meeting the changing needs of the individual), sipIT, to promote fluid consumption in patients with kidney stones. During this intervention, 31 subjects with a history of kidney stones used digital tools (H2Opal connected water bottle, H2OPal mobile app, and a Fitbit smartwatch app [24] for sip gesture detection) for three months. Subjects also completed measures on barriers to adherence, usability, learnability, satisfaction, and success of the sipIT tools. The authors found that the sipIT digital tools were technically feasible and acceptable and reduced barriers for fluid intake guideline adherence [23].

In another study using a "shove" approach via a bottle that overflows if the user hasn't consumed enough, Beddoe and colleagues [25] found that while only 59% of the 24 participants would use the bottle in their daily lives, 92% believed it was effective in promoting healthy hydration behaviors. Also, due to the overflow nature of the bottle, 84% were concerned to leave it on their desk or workspace for fear of a leak, which may lead to disengagement. Lastly,

Cohen and colleagues [26] aimed to validate four commercially available smart bottles in terms of performance (i.e., number of missed sip recordings, mean error, sip mean percent error, mean absolute error, and cumulative mean percent error) and functionality. The study included two phases: 1) a controlled sip volume phase and 2) a free-living phase where a single user drank from the bottle throughout the day as they normally would [26]. The authors concluded that three of the four bottles tested were accurate, easily calibrated, and were consistent over time [26]. However, there is scant research on pregnant women's knowledge of hydration recommendations, behaviors taken to stay hydrated, and their willingness to engage in using digital tools (e.g., smart water bottle, phone app) to promote adequate water intake. The use of digital tools may improve the accuracy of measuring water intake during pregnancy for researchers and participants to effectively promote adequate hydration in the prenatal period.

Thus, to address this gap, the goals of this paper were to describe knowledge of hydration guidelines and water intake behaviors and identify habits and barriers of using digital tools to maintain adequate hydration among pregnant women. Based on previous research [22], it is hypothesized that pregnant women are unaware of hydration recommendations and are not accurately tracking their hydration but will be open and willing to use a smart connected water bottle during pregnancy. It is also hypothesized that knowledge of hydration recommendations and their likelihood/willingness will differ by trimester such that women early in their pregnancies (1st and 2nd trimesters) may be more knowledgeable about hydration recommendations and may be more willing to use a smart connected water bottle compared to women later in their pregnancies (3rd trimester) [16].

## Methods

### Ethics statement

This study was approved by the Pennsylvania State University Institutional Review Board (STUDY00016174). Formal written consent was provided by each participant prior to completing the survey.

### Research participants

Pregnant women were recruited via social media (i.e., Facebook, Twitter, and Instagram), flyers posted in community sites and obstetrics and gynecology clinics, and University research recruitment websites from December 2020 to March 2022. The study was advertised across the United States. Interested women would contact the study team and the team would send a screening survey via REDCap [27,28] to determine eligibility. Women were eligible if they were: currently pregnant, 18 years or older, and able to read and respond in English. Each woman was entered into a monthly drawing for a $50 gift card for completing the survey.

### Study design

If women met the eligibility criteria, they were sent via email, a link to access the consent form and a one-time online survey using the secure data collection instrument REDCap [27,28]. This survey examined their knowledge of prenatal hydration behaviors and acceptability of digital tools (i.e., smartphone app and connected water bottle that tracks fluid consumption and sends reminders to drink throughout the day).

### Research Instrument

The survey was developed based on prior studies to examine formative research, specific validated surveys, and surveys that were adapted to be specific to pregnancy [22,29] (see S1

Questionnaire). One open ended question was asked to understand knowledge of water intake guidelines: "Do you know how much water you should be drinking in a day (in ounces?)" Ounces were then converted to liters. The answers were then dichotomized as: Less than 2.4 L and Greater than 2.4 L. These dichotomized cut-offs were used based on the recommendations the Institute of Medicines Dietary Reference manual [7], which states that pregnant women ages 19–40 years old should obtain approximately 2.3 L of their total daily water intake from beverages, including drinking water [7]. Four questions were asked to understand general behaviors for water intake and fluid consumption: 1) "Do you keep tracking of your daily fluid consumption and if yes, describe the strategies." 2) "What is your preferred beverage of choice" with responses including: Water/Flavored Water, Tea/Iced Tea, Juice, Soda, Coffee, and Other; 3) "Over the past week, how successful have you been at meeting fluid intake guidelines?" with response options ranging from 1-Rarely Successful (0–4 days) to 5- Always Successful (7 days); and 4) "How frequently do you eat food with high water content?" with response options ranging from 1–1 to4 days/week to 3- Multiple times/day.

Four questions were asked to understand habit strength towards drinking water using a 1–3 Likert scale with responses ranging from 1- Disagree to 3- Agree: 1) "Drinking water is something I do automatically."; 2) "Drinking water is something I do without having to consciously remember."; 3) "Drinking water is something I do without thinking."; and 4) "Drinking water is something I start doing before I realize I'm doing it." [23]. One question with 16 pregnancy-specific response options was asked to identify barriers to meeting water intake recommendations: 1) "Which of the following have been barriers to meeting fluid intake guidelines?" Barrier options included: I am not thirsty enough, I forget to drink, It is a hassle to carry around a water bottle, I have to urinate too frequently if I drink that much, It' hard to drink enough at work, I don't like the taste of water, I am not aware of the need to drink more, It makes me feel bloated, Fluid is not easily available at work, I feel nauseous, It keeps me up at night, It is painful to drink that much with the baby pushing down on my bladder, I have a small bladder, I have acid reflux and it is uncomfortable, I have morning sickness, Other. Two Yes/No questions were asked to understand digital tool use: 1) "Have you ever installed an app on your phone or tablet to help you increase fluid consumption (hydration)?" and 2) "Have you ever owned a connected water bottle that tracks or provides reminders about fluid consumption? This water bottle connects to your phone via an app and tracks how much you drink throughout the day based on the weight (volume) of the water/fluid."

Three Likert-scale questions were asked to understand the likelihood/willingness to use digital tools to promote hydration: 1) "How interested would you be in using a new smartphone application or device to aid in meeting the fluid consumption recommendations of 81 ounces per day?" with response options ranging from 1- Uninterested to 3- Interested; 2) "How likely is a smartphone application or device to help you meet the fluid consumption recommendations of 8 ounces per day?"; and 3) "How likely would a connected water bottle to measure/monitor volume of fluids consumed (This water bottle will be connected to your phone via an app to track your water consumption throughout the day)?" with response options ranging from 1- Unlikely to 3- Likely.

## Data collection and analysis

All surveys were completed in the participant's own home and own device via a REDCap survey link. Once the surveys were completed, data was downloaded from the secure survey database by the study staff, into IBM SPSS 28.0.1.0 [30]. Study staff cleaned the data to designate and remove any outliers. Descriptive statistics were used to examine means, standard deviations, frequencies, Mann-Whitney U Tests, and Chi-squared tests were used to determine

significant group differences between early (1$^{st}$ and 2$^{nd}$ trimesters) vs late in pregnancy (3$^{rd}$ trimester), regarding hydration behaviors and acceptability of the digital tools using IBM SPSS 28.0.1.0 [30]. Given pilot exploratory nature of this study, study size was powered on hydration recommendation knowledge. The survey remained opened for 6 months and closed after the 6 month time period.

## Results

### Participant characteristics

A total of $N$ = 404 pregnant women were contacted and provided responses to the survey. Due to a 34% completion rate, only a sample of N = 137 responses were included in the analyses. The majority of women were recruited through social media (50.4%) and from across the United States. Outside of social media, 31.4% of women heard about the study from other internet sources (Google, Research Match, and email), 12.4% heard about the study via word of mouth, and 8.8% saw a flyer at their provider's office. The survey link was included on all social media posts allowing direct access and ability to complete the survey easily and the link was shared easily. A low completion rate could be due to starting the survey and no longer wanting to participate due to various factors such as lack of compensation, the length of the survey, and time constraints. Their mean age was 30.9 years (± 24.3) and had a mean gestational age of 20.9 (± 9.7) weeks gestation. Most women were White (62.8%), employed full time (37.2%), had a bachelor's and/or graduate/professional degree (40.1%), and an annual income of less than $40,000 (74.4%). **Table 1**.

**Knowledge of Water Intake Guidelines.** When asked about how much water pregnant women are recommended to drink each day, most women overall (47%) reported less than 2.4 L, 37% reported greater than 2.4 L, and 16% reported to not know at all. Further exploration by early vs late pregnancy showed that most women in early pregnancy (64%) reported less than 2.4 L, 18% reported greater than 2.4 L, and 18% did not know. Women in late pregnancy reported less than 2.4 L (45%), greater than 2.4 L (41%), and did not know (14%). Women in early pregnancy reported significantly lower recommended values of water/day (1.5 L) compared to women in late pregnancy (2.4 L; U = 666.5, $n_1$ = 15, $n_2$ = 60; $p$ = 0.004).

**Water Intake and Fluid Consumption Behaviors.** When asked about their preferred choice of beverage, 70% of women reported that water/flavored water, 17% reported juice, 9% reported tea/iced tea, 2% reported soda, 1.5% reported coffee, and .75% reported Other. When asked how successful they are at meeting the fluid consumption recommendation each day, most women (67%) reported only being successful in meeting the fluid consumption recommendations 0–4 days/week, 24% were often successful (5–6 days/week), and 8% were always successful (7 days/week). When asked how often women eat foods high in water (i.e., fruits, vegetables), the majority (83%) reported eating foods high in water multiple times a day over the course of a week, 9% reported eating high water foods 4–6 days/week, and 8% reported eating high water foods 0–3 days/week.

**Habit Strength Toward Drinking Water.** When asked if drinking water is something they do automatically, most women (82%) agreed, 9% disagreed, and 9% neither agreed nor disagreed (**Fig 1A**). When asked if drinking water is something they do without thinking, most women (77%) agreed, 13% disagreed, and 10% neither agreed nor disagreed (**Fig 1B**). When asked if drinking water is something they do without having to consciously remember, most women (75%) agreed, 11% disagreed, and 14% neither agreed nor disagreed (**Fig 1C**). When asked if drinking water was something they start doing before they realize they're doing it, most women agreed (65%) agreed, 12% disagreed, and 23% neither agreed nor disagreed (**Fig 1D**). However, when asked if they currently track their fluid intake, 59% of women

**Table 1. Participant demographics of pregnant women (*N* = 137).**

|  | Mean ± SD | N (%) |
|---|---|---|
| **Age** | 30.9 ± 24.3 years | - |
| **Gestational Age** | 20.9 ± 9.7 weeks | - |
| **Race/Ethnicity** | - |  |
| White | - | 86 (62.8) |
| American Indian/Alaskan Native | - | 29 (21.2) |
| African American/Black | - | 4 (2.9) |
| Asian | - | 7 (5.1) |
| Other | - | 3 (2.9) |
| Native Hawaiian or Other Pacific Islander | - | 3 (2.9) |
| Two or more | - | 1 (0.7) |
| **Employment** | - |  |
| Full-Time/Self-Employed | - | 51(37.2) |
| Part-Time | - | 39 (28.5) |
| Unemployed | - | 36 (26.3) |
| Student | - | 4 (2.9) |
| Retired |  | 2 (2.2) |
| **Education** | - |  |
| Less than High School | - | 18 (12.4) |
| High school/GED | - | 17(12.4) |
| Some College, no degree | - | 16 (11.7) |
| Trade/Technical School | - | 17 (12.4) |
| Associate degree | - | 10 (7.3) |
| Bachelor's degree | - | 31 (22.6) |
| Graduate/Professional degree | - | 24 (17.5) |
| **Family Income** | - |  |
| <$40,000 | - | 102 (74.4) |
| >$40,000 | - | 30 (21.9) |
| Other | - | 1 (.7) |

reported no, 31% reported yes, and 10% were unsure (**Fig 2A**). Further exploration by early vs late pregnancy showed that women in early (58%) and late (61%) pregnancy reported no, 29% (early) and 31% (late) reported yes, and 13% (early) and 8% (late) were unsure (**Fig 2B**). There were no significant differences between early vs late pregnancy (p's > 0.05). When asked their current tracking method, many women (31%) reported they do not but 26% reported they count their water bottles/how many times they fill it. Other responses for this question can be found in **Table 2**.

**Barriers to Meeting Fluid Intake Recommendations.** Women were asked about major barriers that keep them from meeting the recommended fluid intake and the most frequently reported answers included: forgetting to drink (47%), not feeling thirsty (47%), causing increased urination (33%), feeling it is a hassle (24%), and makes them feel bloated and nauseous (20%). A bar graph of these and other responses can be found in **Fig 3**.

**Likelihood/Willingness to Use Digital Tools.** When asked if they have ever installed an app on their phone to help increase fluid consumption, most women (56%) reported no, 41% reported yes, and 3% were unsure. Further exploration by early vs late pregnancy showed that most women in early pregnancy (56%) reported yes, 36% reported no, and 8% were unsure. Women in late pregnancy reported yes (37%) and reported no (63%). When asked if they've ever owned a smart connected water bottle that tracks or provides reminders about fluid

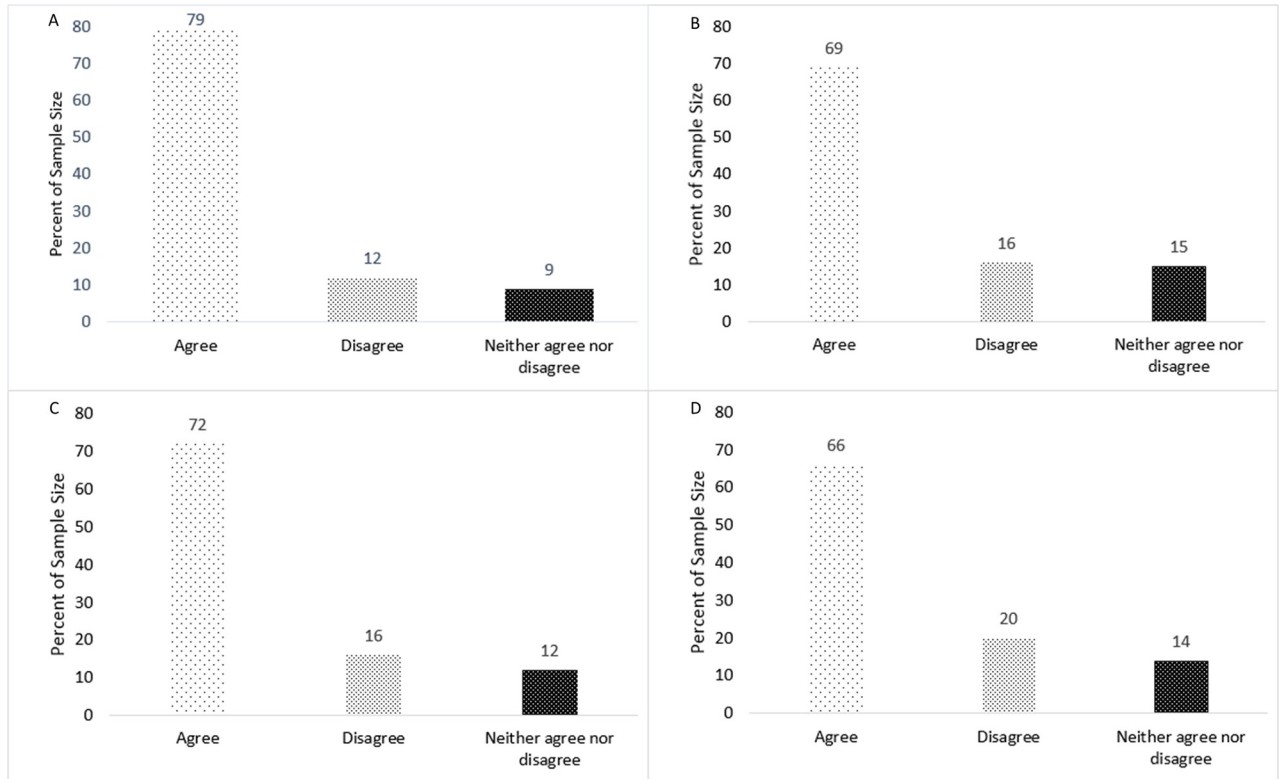

**Fig 1.** A. Drinking water is something I do automatically (*N* = 130). B. Drinking water is something I do without thinking (*N* = 125). C. Drinking water is something I do without having to consciously remember (*N* = 125). D. Drinking water is something I start doing before I realize I'm doing it (*N* = 124).

consumption, most women (61%) responded no, 36% reported yes, and 3% were unsure. Further exploration by early vs late pregnancy showed that most women in early and late pregnancy reported no (56% and 66%, respectively), 44% and 33% reported yes, respectively, and only 1% in late pregnancy were unsure. When asked how interested they would be in using a new smartphone application or device to aid in meeting the fluid consumption

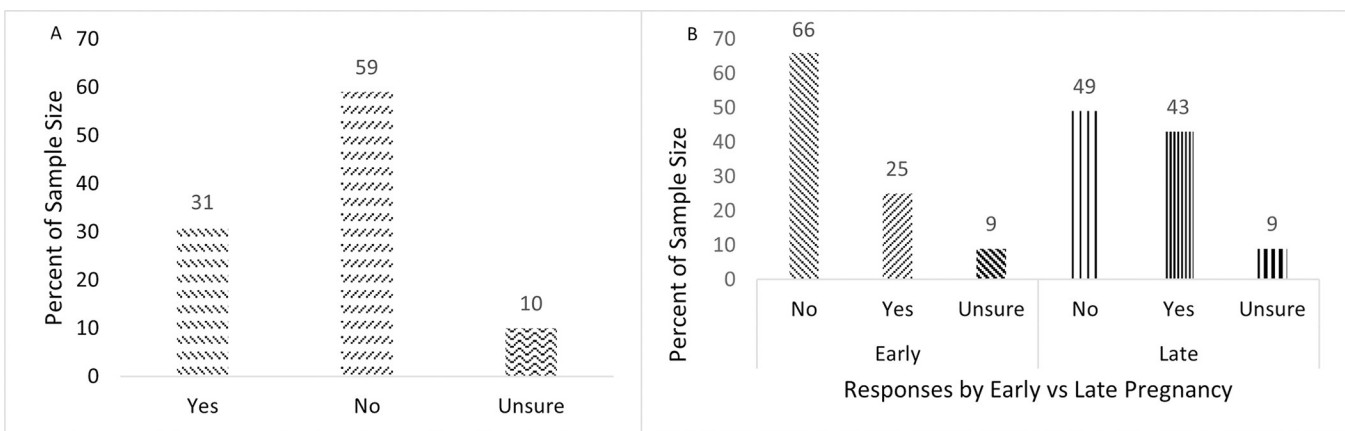

**Fig 2.** A. Do you keep track of your daily fluid consumption? Overall sample (*N* = 124). B. Do you keep track of your daily fluid consumption? By early vs late pregnancy (early pregnancy *n* = 24, late pregnancy *n* = 91).

**Table 2. Responses to Hydration Knowledge, Behaviors, and Digital Tool Feasibility Questions (*N* = 135).**

| Please describe the strategies you CURRENTLY use to track your fluid consumption: | | |
|---|---|---|
| **Response** | ***N*** | **%** |
| None | 42 | 31 |
| Count water bottles | 35 | 26 |
| Drink consistently throughout the day | 13 | 10 |
| Drink when pregnancy symptoms come up | 12 | 9 |
| Set an ounce goal | 8 | 6 |
| Notebook | 8 | 6 |
| Tracking app | 7 | 5 |
| Drink preferable liquids | 6 | 4 |
| Drink with meals | 2 | 1.5 |
| Electronic scale | 2 | 1.5 |

recommendations each day, most women (69%) reported they would be interested, 21% reported to be uninterested, and 10% were neither interested nor uninterested. Further exploration by early vs late pregnancy showed that most women in early and late pregnancy would be interested (84% and 67%, respectively), 12% and 22% would be uninterested, respectively, and 4% and 11%, respectively were neither interested nor uninterested. When asked how likely they would use a smartphone application or device to aid in meeting the fluid consumption recommendations each day, most women (68%) responded likely, 20% reported unlikely, and 12% reported neither likely nor unlikely. Further exploration by early vs late pregnancy showed that most women in early and late pregnancy responded likely (88% and 64%,

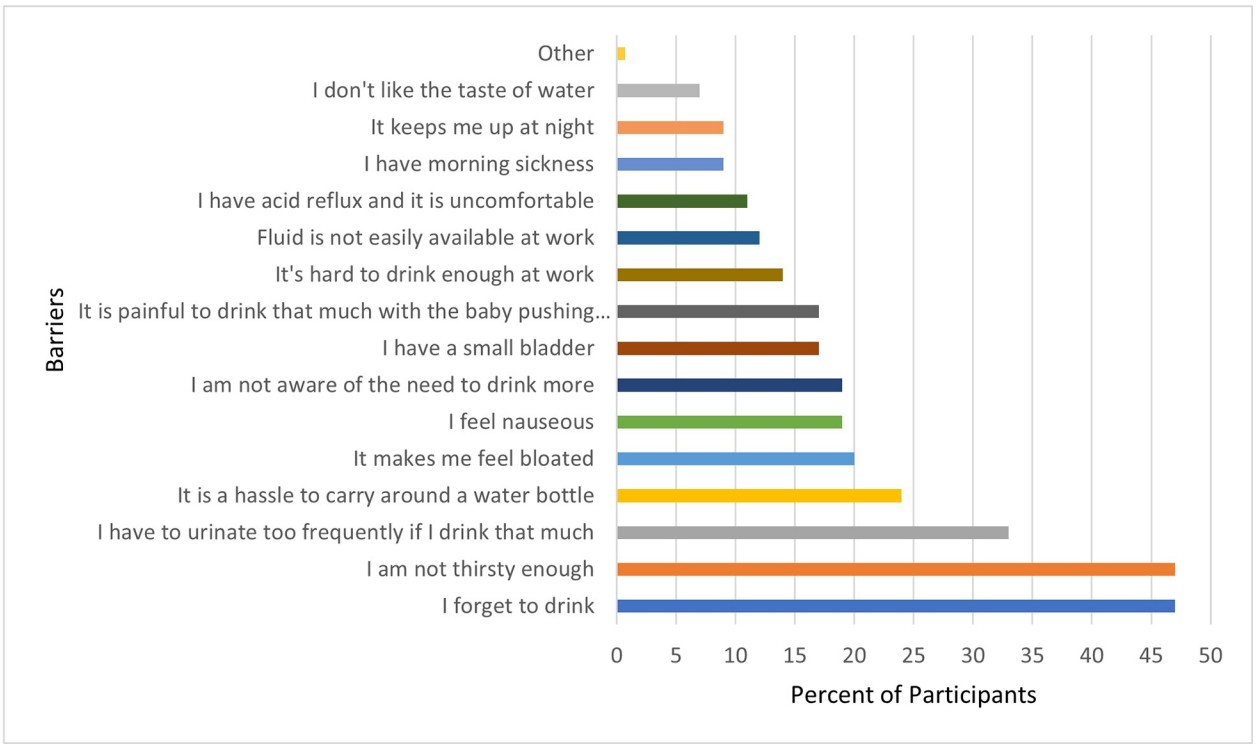

**Fig 3. Barriers to meeting fluid intake guidelines (*N* = 137).**

respectively), 8% and 23% reported unlikely, respectively, and 4% and 13%, respectively reported neither likely nor unlikely. When asked how likely a smartphone application or device would aid in meeting the fluid consumption recommendations each day, most women (67%) responded likely, 20% responded unlikely, and 13% were unsure. Further exploration by early vs late pregnancy showed that most women in early and late pregnancy responded likely (71%, both), 25% and 19% reported unlikely, respectively, and 4% and 10%, respectively were unsure. There were no significant differences between early vs late pregnancy for any of the questions (p's > 0.05).

## Discussion

This is one of the first studies to our knowledge to describe prenatal hydration knowledge, habits, behaviors, and barriers as well as identify women's willingness to use digital tools to maintain adequate hydration during pregnancy. Overall, most women were unaware of the fluid consumption recommendations, were not successful in meeting the recommendations, and do not regularly track their fluid consumption. Women later in pregnancy knew of and were exceeding the hydration recommendations compared to women early in pregnancy. However, we also found that most women owned a smartphone or tablet/computer and reported that they would be interested in and likely use a smart, connected water bottle to help increase their fluid consumption. These findings highlight that pregnant women may want to use these useful strategies to increase knowledge, decrease barriers, and maintain adequate hydration over pregnancy, particularly before conception and early in pregnancy to improve hydration behaviors throughout the entire pregnancy. The results are described in more detail below.

Consistent with previous findings [12,31], we also observed that women lacked knowledge about guidelines, were not adequately meeting the hydration recommendations, and were not regularly tracking their water intake. Similarly, when examining hydration behaviors in former kidney stone patients, the majority of pregnant women forgot to drink or didn't feel thirsty [28]. However, we found that most women reported they drink water throughout the day, which is in direct contrast to the majority reporting they forget to drink or didn't feel thirsty. Our study findings suggest there is a gap between women's knowledge and their actual hydration behaviors, and perceived behavior highlights the need to promote adequate hydration behaviors during pregnancy. Our findings also highlight the need for education and increased knowledge prior to conception and early in the pregnancy to continue healthy hydration behaviors throughout the entire pregnancy and into postpartum as many women were not meeting the hydration recommendations. This gap may increase the risks for adverse pregnancy and postpartum outcomes (i.e., excessive gestational weight gain, hypertensive disorders, preterm births, and gestational diabetes [32]), which increases the risk for maternal mortality. Thus, there is a need for strategies to promote knowledge and adequate intake recommendations and strategies during pregnancy.

The majority of women stated they own a smartphone or tablet but have never used an app or smart connected water bottle to track their water consumption. The development and integration of digital tools to monitor various lifestyle behaviors has been on the rise over the last two decades. For example, Fitbit has developed and disseminated multiple wearable activity and sleep tracking devices and WiFi/Bluetooth connected weight scales along with their smartphone app [24]. The app has various features that tracks and displays data on physical activity and sleep patterns but also integrates eating and hydration behaviors [24]. However, for the app to track these behaviors, the user has to manually enter the information. As stated above, many barriers include forgetting to drink and not feeling thirsty. If women are forgetting to

drink and aren't feeling thirsty, they most likely will not utilize the app to track their behaviors if they need to manually input the information. Many women had reported using and counting water bottles as their tracking method, which could be a reason for the wide gap in knowledge, perceived intake, and actual intake. It has also been found that there is under- and over-reporting when pregnant women self-report their dietary behaviors, which decreases the accuracy of their behaviors and intake. More advanced technology that takes away the self-report nature of hydration tracking may be useful, particularly when used during pregnancy and provides a more hands-off approach when counseling women on their hydration needs. However, there are other novel technologies outside of water bottles that may be useful such as wrist-worn sensors that monitor sip gestures by hand/arm movement that can estimate fluid intake amount passively [33,34]. These digital tools may be more advantageous to use in tandem, in order to gather more behavior data and promote hydration.

Since women are already using water bottles to drink water, the use of a smart water bottle would be an easy transition as well as a more enjoyable experience. Since the water bottle tracks how much they have been drinking throughout the day, sends reminders to drink, and has the capabilities to set goals and motivate women to drink more water, this could promote the increase in their confidence to meet goals and recommendations by increasing their fluid intake in a virtual, free-living settings. More importantly, to reduce the risks of adverse pregnancy and postpartum outcomes and maternal mortality, using this water bottle in conjunction with other digital tools, such as wearable sensors, provides insight on and the promotion of hydration behaviors without being too intrusive or life disrupting when preparing to delivery or caring for an infant.

## Strengths and limitations

There were several strengths of this study, one of which was that it was one of the first to understand pregnant women's thoughts, beliefs, preferences, and needs regarding hydration, hydration behaviors, and the use of digital tools overall but also based on trimester. Previous research has focused on associations of hydration levels on pregnancy outcomes [10,35], measuring hydration levels during pregnancy [11,32] and intravenous hydration [36], while this study focused on understanding how researchers can promote hydration behaviors during pregnancy. These findings highlight the major knowledge and research gaps when it comes to hydration behaviors during pregnancy. More research is needed to fully understand the novelty of a smart connected water bottle and other digital tools when trying to promote hydration behaviors. However, this study was not without limitations. The study sample was mostly a homogenous sample of white, fully employed, middle income women and results may not be generalizable to other groups within the pregnant population. Replication of this study is needed for women with more diverse backgrounds, as Black and Hispanic women are less likely to drink water, to understand their hydration behaviors and to determine if the use of digital tools would be as well received or useful [37]. Further, water recommendations are not a one-size fits all. Greater body composition and body mass, living in hotter temperatures and engaging in physical activity all increase water needs and thus some pregnant women may need more or less than the AI recs depending on their specific situation [11]. The data were cross-sectional and do not accurately portray how thoughts, beliefs, and preferences might change over the course of pregnancy. The data were also self-reported and may have introduced bias and/or under- or over-reporting of their water intake so there may not have been accurate values of water consumption. Lastly, there are varied ranges in the literature for recommended water intake values of 1.9–2.8 L (64–96 ounces) [9]. This study used a dichotomous cutoff of 2.3 L (81 ounces) [7] to classify meeting vs not meeting hydration

recommendations for feasibility purposes but future work should explore variations in water intake as the literature suggests [7,9].

## Conclusions

The findings from this study suggest that pregnant women are unaware of prenatal hydration recommendations, potentially the adverse pregnancy and infant outcomes under-hydration may cause, and do not accurately track their fluid intake. However, women in later pregnancy knew of and were exceeding hydration recommendations compared to women in early pregnancy. This important finding highlights the need for increased education and knowledge during the preconception and early pregnancy time frames. This will allow for healthy hydration behaviors to be sustained throughout pregnancy and postpartum. The study findings also suggest that women are interested in and feel that the utilization of digital tools (e.g., smart water bottle and app) would be useful in increasing fluid consumption during pregnancy. By providing reminders to drink, this will help overcome the barrier of forgetting to drink. However, there weren't any significant differences between early vs late pregnancy. The use of the water bottle and app will allow for women to self-monitor, get regular feedback on their behaviors, and create challenges to increase their motivation to meet their goals and recommendations so they start and continue to drink water, even if they don't necessarily feel thirsty. However, many hydration recommendations, education, and digital tools, are not pregnancy specific. More research and development of recommendation, educational content, and digital tools that are pregnancy specific are needed. As such, education on prenatal fluid consumption recommendations, the benefits of staying hydrated, setting goals and tracking using digital tools, can be built into hydration promotion interventions, particularly during preconception and early pregnancy. By using digital tools to promote and increase hydration behaviors during pregnancy, the risks associated with adverse pregnancy outcomes may be lessened.

## Supporting information

**S1 Questionnaire. Demographics & health history.**
(DOCX)

## Acknowledgments

The authors would like to acknowledge the assistance of the Exercise Psychology Laboratory Research Assistants at the Pennsylvania State (University Park campus) who assisted with data collection and analyses for this study.

The content is the sole responsibility of the authors and does not necessarily represent the official views of the NIH/NIDDK, the Pennsylvania State University or NCATS.

## Author Contributions

**Conceptualization:** Abigail M. Pauley, Asher Y. Rosinger, Jennifer S. Savage, David E. Conroy, Danielle Symons Downs.

**Data curation:** Abigail M. Pauley.

**Formal analysis:** Abigail M. Pauley.

**Investigation:** Abigail M. Pauley.

**Methodology:** Abigail M. Pauley, Asher Y. Rosinger, Jennifer S. Savage, David E. Conroy, Danielle Symons Downs.

**Project administration:** Abigail M. Pauley, Danielle Symons Downs.

**Resources:** Asher Y. Rosinger, Jennifer S. Savage, David E. Conroy, Danielle Symons Downs.

**Supervision:** Asher Y. Rosinger, Jennifer S. Savage, David E. Conroy, Danielle Symons Downs.

**Validation:** Asher Y. Rosinger, Jennifer S. Savage, David E. Conroy, Danielle Symons Downs.

**Visualization:** Abigail M. Pauley.

**Writing – original draft:** Abigail M. Pauley, Asher Y. Rosinger, Jennifer S. Savage, David E. Conroy, Danielle Symons Downs.

**Writing – review & editing:** Abigail M. Pauley, Asher Y. Rosinger, Jennifer S. Savage, David E. Conroy, Danielle Symons Downs.

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
