## [Decision Letter · Decision Letter 0]

17 Jul 2023

PDIG-D-23-00237

Every Sip Counts: Understanding hydration behaviors and user-acceptability of digital tools to promote adequate intake during pregnancy

PLOS Digital Health

Dear Dr. Pauley,

Thank you for submitting your manuscript to PLOS Digital Health. After careful consideration, we feel that it has merit but does not fully meet PLOS Digital Health's publication criteria as it currently stands. Therefore, we invite you to submit a revised version of the manuscript that addresses the points raised during the review process.

Please submit your revised manuscript within 60 days Sep 15 2023 11:59PM. If you will need more time than this to complete your revisions, please reply to this message or contact the journal office at digitalhealth@plos.org. Please include the following items when submitting your revised manuscript:

We look forward to receiving your revised manuscript.

Kind regards,

Haleh Ayatollahi

Section Editor

PLOS Digital Health

Journal Requirements:

1. Please send a completed 'Competing Interests' statement, including any COIs declared by your co-authors. If you have no competing interests to declare, please state "The authors have declared that no competing interests exist". Otherwise please declare all competing interests beginning with twhe statement "I have read the journal's policy and the authors of this manuscript have the following competing interests:"

b. If any authors received a salary from any of your funders, please state which authors and which funders.

3. In the online submission form, you indicated that "The data can be accessed through contacting the corresponding author". All PLOS journals now require all data underlying the findings described in their manuscript to be freely available to other researchers, either 1. In a public repository, 2. Within the manuscript itself, or 3. Uploaded as supplementary information.

Additional Editor Comments (if provided):

The manuscript was interesting. Please address the following comments in your revision.

1- Please follow the journal instructions for manuscript preparation. I assume the abstract should be unstructured.

2- Please add appropriate keywords (preferably selected based on the MeSH terms) after the abstract.

3- Please use appropriate subheadings in your methods section (study design, research participants, research instrument, data collection and analysis, etc.) to organize and improve readability of this section.

4- Please add the questionnaire as an Appendix.

5- Please ensure that the data for all questions have been presented in the results section either via Table or Figure.

6- References older than 10 years ago need to be updated.

Reviewers' comments:

Reviewer's Responses to Questions

**Comments to the Author**

1. Does this manuscript meet PLOS Digital Health’s publication criteria? Is the manuscript technically sound, and do the data support the conclusions? The manuscript must describe methodologically and ethically rigorous research with conclusions that are appropriately drawn based on the data presented.

Reviewer #1: Partly

Reviewer #2: Yes

Reviewer #3: Partly

2. Has the statistical analysis been performed appropriately and rigorously?

Reviewer #1: N/A

Reviewer #2: Yes

Reviewer #3: Yes

3. Have the authors made all data underlying the findings in their manuscript fully available (please refer to the Data Availability Statement at the start of the manuscript PDF file)?

Reviewer #1: No

Reviewer #2: Yes

Reviewer #3: No

4. Is the manuscript presented in an intelligible fashion and written in standard English?

Reviewer #1: Yes

Reviewer #2: Yes

Reviewer #3: Yes

5. Review Comments to the Author

Reviewer #1: For this journal, it is not allowed to have only one person as the corresponding author for your data. In the future, if you are not allowed to share your data in a de-identified fashion, you should put the ethics committee that oversees data sharing. Please refer to the guidelines for PLoS Digital Health here that might be helpful for future submissions: https://journals.plos.org/digitalhealth/s/data-availability You can then update your data availability statement with the committee that oversees data sharing up at University Park. 

Also, in the introduction, you cited a Chinese study, but the majority of your respondents are white. You might want to put that article in the discussion section if you are aiming to broaden your reach for future research. 

Do not use contractions unless you are capturing responses from your participants, as it is considered not scholarly. 

This article is probably a better fit for International Journal of Behavioral Nutrition and Physical Activity or Journal of Nutrition.

Reviewer #2: Interesting paper! I have a few comments and suggestion on how to improve the paper. 

Comments to PDIG-D-23-00237

Line 87: Is it in the guidelines that pregnant women are encouraged to “track their water intake”? Please show me a reference.

Line 88: “New mothers” – do you mean first time being pregnant? Or first time giving birth? Please show me a reference for the statement on greater risk for maternal and infant mortality when not hydrating enough.

Line 94: I don’t understand what is written in the parathesis: N=300, 42% - is 42% 300 people, or is it 42% of the 300 people? The same for the next line on 14%

Line 99: Please use L or ounces throughout the text. I prefer L. 

Line 106: 81 ounces – see previous comment

Line 107: MyFitnessPal app – I need a reference, or a more explanation for what this is and where it was produced. 

Line 117: Is it possible to have a picture of a “smart water bottle”? I have never seen one.

Line 123: What is a “Just-in-time adaptive intervention”? Could you please elaborate?

Line 129: I do not understand why anyone want a bottle that overflows when the consumer has consumed enough if you want to increase water intake? If I had a bottle like that, I would just stop drinking to avoid the overflow??

Line 148-150: Why do women in later pregnancy have a decreased knowledge of hydration recommendation? 

Line 153: You recruited for more than 1 year. Why did you only include 137? It seems like you have tried to reach out in many ways, but why did you end up with such small and monotonous sample size?

Line 163: What if you want to choose exactly 81 ounces? 

Line 168: Do the pregnant women who answers this survey know what “food with high water content” is? Are there mentioned examples in the survey?

Line 199: Why was the survey for 6 months? So the women could change their answers? Or? Why was this needed?

Line 202: Why did so few complete the survey? Was it very long? Who was it difficult to complete for? How did they differ from those who was included?

Line 211: 8ounces or 81ounces?

Line 216: Is “preferred choice of beverage” referred to want they drink the most or what the like the most? In my opinion it could be misunderstood?

Line 223: The sentence ends with “and” – what did you want to say more?

Figure 1a to Figure 1d (Line 224 and forth): What is the purpose of asking the same question in 4 different ways?

Line 273: If you want to state that “pregnant women may benefit from useful strategies to increase knowledge…”, I think, you need to show that those who did track their fluid intake, had a higher fluid intake than those who did not. 

Line 276: So you did investigate for a pattern in the answers? Those who did not meet hydration recommendation where they also those who did not know the hydration recommendation and did not track their fluid intake and so forth? Or? Please state this in the method section and mention in the result section.

Line 281: I do not see how your results “suggest there is a gap between women’s knowledge and their actual hydration behaviors” if the sentence before should be the reason for this statement. In the sentence before you present that pregnant women do not feel thirsty or forgot to drink and also that they drink water throughout the day – nothing about knowledge??

Line 284-285: Please put in a reference 

Line 289: Fitbit – please give a reference (company, country, year)

Line 302: “wrist-worn sensors” – why did you not include this in the survey if you know it exists?

Line 321: Nice discussion of limitations of the study!

Additional: Did you compare answers of the survey and patient characteristics? Did you find any pattern when examining this? Maybe this could improve the planning of future studies within this field?

Reviewer #3: Every Sip Counts: Understanding hydration behaviors and user-acceptability of digital tools to promote adequate intake during pregnancy

I congratulate the author for this innovative study. The author attempted a new aspect regarding hydration during pregnancy; the assessment of participants knowledge of recommendations and the use of digital tools to increase water intake. Moreover, the author has taken care of handling missed data. The abstract is written in the way it gives useful information for the paper.

Introduction

I appreciate how the introduction discuss previous evidence related to both hydration behavior and use of digital tools to track water intake among pregnant women. Searched evidences informed the gap in the field which make clear the problem the author wants to address and the rationale behind. 

The author emphasizes (L147-150) on the difference in knowledge of hydration recommendations and willingness of using digital tool according to the age of pregnancy. However, this is not reflected in the title. 

Materials and methods

• L153: To avoid a possible confusion, I was wondering if it is not wise to fully write OB/GYN even if it is known what it stands for. Why flyers were not also posted in antenatal care facilities where majority of pregnant women frequently visit in? I assume that OB/GYN clinics are rarely visited and especially for a serious issue or a referral case. 

• L154: It not clearly shown how participant were screened against eligibility criteria. I would also like to know if an interested participant had to request for the link to access the online survey. 

• The study setting is not defined. Was the study conducted in a particular community?

• L156-157: I think that a participating woman was not aware of the $50 gift card before otherwise it could be a source of bias. 

• I think it would be good idea to ask question about types of apps used to track water intake because different apps may have different effect on participants acceptability; some may be more acceptable than others.

• Nothing is mentioned concerning informing to participants about the study and getting their consent.

• Why was the survey remained opened for exactly 6 months? any particular reason/explanation for that?

• I would be better to state how data were managed for analysis to ease the reader understanding.

Results

• Table 1 describing participant demographics of pregnant women shows that two women were retired and I doubt that a woman can be pregnant when she has reached an age of retirement…

• L211: a typo “8ounces”

• Table two show that only 7 (5%) participants were using a tracking app what seems to be in contrast with what is reported concerning interest in use of new app (L247-252)

Discussion

• L271: I hope it is an assumption that most women own a smartphone or a tablet/computer. I have not seen it in the results 

• I recommend the author to search more other evidences to discuss his findings on the two aspects; knowledge and use of digital tools in tracking water intake. 

Conclusion

• The manuscript describes hydration behaviors and user-acceptability of digital tools to promote adequate intake during pregnancy.

• The result section is heavy reporting much data. The author should report key findings to guide the conclusion

Congratulations to the authors and happy to read an improved version in the near future!

6. PLOS authors have the option to publish the peer review history of their article (what does this mean?). If published, this will include your full peer review and any attached files.

**Do you want your identity to be public for this peer review?** For information about this choice, including consent withdrawal, please see our <a href="https://www.plos.org/privacy

---

## [Decision Letter · Decision Letter 1]

14 Dec 2023

PDIG-D-23-00237R1

Every Sip Counts: Understanding hydration behaviors and user-acceptability of digital tools to promote adequate intake during early and late pregnancy

PLOS Digital Health

Dear Dr. Pauley,

Thank you for submitting your manuscript to PLOS Digital Health. After careful consideration, we feel that it has merit but does not fully meet PLOS Digital Health's publication criteria as it currently stands. Therefore, we invite you to submit a revised version of the manuscript that addresses the points raised during the review process.

Please submit your revised manuscript within 60 days Feb 12 2024 11:59PM. If you will need more time than this to complete your revisions, please reply to this message or contact the journal office at digitalhealth@plos.org. Please include the following items when submitting your revised manuscript:

We look forward to receiving your revised manuscript.

Kind regards,

Haleh Ayatollahi

Section Editor

PLOS Digital Health

Journal Requirements:

1. Please provide separate figure files in .tif or .eps format only and remove any figures embedded in your manuscript file. Please also ensure that all files are under our size limit of 10MB.

2. Please insert an Ethics Statement at the beginning of your Methods section, under a subheading 'Ethics Statement'. It must include:

1) The name(s) of the Institutional Review Board(s) or Ethics Committee(s)

2) The approval number(s), or a statement that approval was granted by the named board(s) 

3) (for human participants/donors) - A statement that formal consent was obtained (must state whether verbal/written) OR the reason consent was not obtained (e.g. anonymity). NOTE: If child participants, the statement must declare that formal consent was obtained from the parent/guardian.

Additional Editor Comments (if provided):

Reviewers' comments:

Reviewer's Responses to Questions

**Comments to the Author**

1. If the authors have adequately addressed your comments raised in a previous round of review and you feel that this manuscript is now acceptable for publication, you may indicate that here to bypass the “Comments to the Author” section, enter your conflict of interest statement in the “Confidential to Editor” section, and submit your "Accept" recommendation.

Reviewer #4: All comments have been addressed

Reviewer #5: (No Response)

2. Does this manuscript meet PLOS Digital Health’s publication criteria? Is the manuscript technically sound, and do the data support the conclusions? The manuscript must describe methodologically and ethically rigorous research with conclusions that are appropriately drawn based on the data presented.

Reviewer #4: Partly

Reviewer #5: Yes

3. Has the statistical analysis been performed appropriately and rigorously?

Reviewer #4: No

Reviewer #5: Yes

4. Have the authors made all data underlying the findings in their manuscript fully available (please refer to the Data Availability Statement at the start of the manuscript PDF file)?

Reviewer #4: No

Reviewer #5: Yes

5. Is the manuscript presented in an intelligible fashion and written in standard English?

Reviewer #4: Yes

Reviewer #5: No

6. Review Comments to the Author

Reviewer #4: Overall, this study attempts to determine whether women understand hydration behaviors and evaluate their user-acceptability of digital tools to promote adequate intake during pregnancy. I commend the authors for undertaking this unique and interesting topic. However, there are some major issues that should be addressed.

1. Line 162, 163: The survey provided an open answer question “Do you know how much water you should be drinking in a day (in ounces?)”, and the answer was dichotomized into less than 81 ounces and greater than 81 ounces. While this serves as a proxy of the water consumption adequacy, the authors should justify why this cutoff was chosen, since recommended water intake is largely based on other metrics including age, BMI, , activity level, comorbidities..etc. The recommended water intake amount during pregnancy according to ACOG on their official site 8 to 12 cups (64 to 96 ounces) (https://www.acog.org/womens-health/experts-and-stories/ask-acog/how-much-water-should-i-drink-during-pregnancy )

2. The citations listed on line 106 regarding the recommended amount of water intake: meet the hydration guidelines of consuming at least 81 ounces of water from ~8-36 weeks’ gestation [15,16]. Both articles did not explicitly state “81 ounces” as the recommended amount of water intake, other citations for justifying the “81 ounce” should be provided.

3. The completion rate of surveys was 34%. Could the authors specify the predominant channel of recruitment for those who completed the survey (patients who were included in the analysis)? Understanding the behavioral profile of participants sourced through diverse platforms such as social media, community flyers, OB/GYN clinics, and University websites could offer nuanced insights.

4. The tables presented in the manuscript are mostly descriptive data. But regarding the statistics of comparing between early and late pregnancy, only 137 samples were analyzed between 2 groups (early vs late), yet the authors chose One-way ANOVA: a parametric test which is used for 3 or more groups for the analysis. It is unlikely that the numbers from the small sample size would be normally distributed. I recommend re-doing the analysis with a non-parametric test for 2 groups (Wilcoxon rank sum test). Moreover, considering the binary nature of most survey items (yes/no), shouldn't the chi-squared test be apt for comparing categorical variables across groups? The omission of this in the methodology is confusing.

Reviewer #5: Some minor errors still need to be corrected: lack of close parenthesis in line 82, and in-text citation formatting in lines 291 and 292. Please follow PLOS guidelines for the citation format.

7. PLOS authors have the option to publish the peer review history of their article (what does this mean?). If published, this will include your full peer review and any attached files.

**Do you want your identity to be public for this peer review?** For information about this choice, including consent withdrawal, please see our Privacy Policy. 

Reviewer #4: No

Reviewer #5: Yes: Arianne Justine T. Obeles, MD

---

## [Decision Letter · Decision Letter 2]

29 Mar 2024

Every Sip Counts: Understanding hydration behaviors and user-acceptability of digital tools to promote adequate intake during early and late pregnancy

PDIG-D-23-00237R2

Dear Ms Pauley,

We are pleased to inform you that your manuscript 'Every Sip Counts: Understanding hydration behaviors and user-acceptability of digital tools to promote adequate intake during early and late pregnancy' has been provisionally accepted for publication in PLOS Digital Health.

Best regards,

Haleh Ayatollahi

Section Editor

PLOS Digital Health

Reviewer Comments (if any, and for reference):

Reviewer's Responses to Questions

**Comments to the Author**

1. If the authors have adequately addressed your comments raised in a previous round of review and you feel that this manuscript is now acceptable for publication, you may indicate that here to bypass the “Comments to the Author” section, enter your conflict of interest statement in the “Confidential to Editor” section, and submit your "Accept" recommendation.

Reviewer #6: All comments have been addressed

Reviewer #7: All comments have been addressed

2. Does this manuscript meet PLOS Digital Health’s publication criteria? Is the manuscript technically sound, and do the data support the conclusions? The manuscript must describe methodologically and ethically rigorous research with conclusions that are appropriately drawn based on the data presented.

Reviewer #6: Partly

Reviewer #7: Yes

3. Has the statistical analysis been performed appropriately and rigorously?

Reviewer #6: N/A

Reviewer #7: Yes

4. Have the authors made all data underlying the findings in their manuscript fully available (please refer to the Data Availability Statement at the start of the manuscript PDF file)?

Reviewer #6: No

Reviewer #7: Yes

5. Is the manuscript presented in an intelligible fashion and written in standard English?

Reviewer #6: Yes

Reviewer #7: Yes

6. Review Comments to the Author

Reviewer #6: The revised manuscript has addressed all the comments and can be a suitable research for highlighting the significance of proper hydration during pregnancy. Although the significance of digital methods has not been found more significant towards the hydartion habit, but using specific digital tools for implementing hydartion practice during pregnancy could atleast make the women aware about the importance of drinking water during those critical time period and prevent the possibilities for occurrence of the abnormalities in the baby and complications during the delivery.

Reviewer #7: Thank you for a very interesting research and manuscript. Going forward, a multilingual tool would be beneficial to be able to reach a more diverse population.

7. PLOS authors have the option to publish the peer review history of their article (what does this mean?). If published, this will include your full peer review and any attached files.

**Do you want your identity to be public for this peer review?** For information about this choice, including consent withdrawal, please see our Privacy Policy.

Reviewer #6: No

Reviewer #7: **Yes: **
